# Fasting mimicking diet as an adjunct to neoadjuvant chemotherapy for breast cancer in the multicentre randomized phase 2 DIRECT trial

Stefanie de Groot[1], Rieneke T. Lugtenberg[1], Danielle Cohen[2], Marij J. P. Welters[1], Ilina Ehsan[1], Maaike P. G. Vreeswijk[3], Vincent T. H. B. M. Smit[2], Hiltje de Graaf[4], Joan B. Heijns[5], Johanneke E. A. Portielje[1,6], Agnes J. van de Wouw[7], Alex L. T. Imholz[8], Lonneke W. Kessels[8], Suzan Vrijaldenhoven[9], Arnold Baars[10], Elma Meershoek-Klein Kranenbarg[11], Marjolijn Duijm-de Carpentier[11], Hein Putter[12], Jacobus J. M. van der Hoeven[1], Johan W. R. Nortier[1], Valter D. Longo[13,14], Hanno Pijl[15], Judith R. Kroep[1 ✉] & Dutch Breast Cancer Research Group (BOOG)[16],*

Short-term fasting protects tumor-bearing mice against the toxic effects of chemotherapy while enhancing therapeutic efficacy. We randomized 131 patients with HER2-negative stage II/III breast cancer, without diabetes and a BMI over 18 kg m$^{-2}$, to receive either a fasting mimicking diet (FMD) or their regular diet for 3 days prior to and during neoadjuvant chemotherapy. Here we show that there was no difference in toxicity between both groups, despite the fact that dexamethasone was omitted in the FMD group. A radiologically complete or partial response occurs more often in patients using the FMD (OR 3.168, $P = 0.039$). Moreover, per-protocol analysis reveals that the Miller&Payne 4/5 pathological response, indicating 90–100% tumor-cell loss, is more likely to occur in patients using the FMD (OR 4.109, $P = 0.016$). Also, the FMD significantly curtails chemotherapy-induced DNA damage in T-lymphocytes. These positive findings encourage further exploration of the benefits of fasting/FMD in cancer therapy. Trial number: NCT02126449.

[1] Department of Medical Oncology, Leiden University Medical Center, P.O. Box 9600, 2300 RC Leiden, The Netherlands. [2] Department of Pathology, Leiden University Medical Center, P.O. Box 9600, 2300 RC Leiden, The Netherlands. [3] Department of Human Genetics, Leiden University Medical Center, P.O. Box 9600, 2300 RC Leiden, The Netherlands. [4] Department of Medical Oncology, Medical center Leeuwarden, P.O. Box 888, 8901 NR Leeuwarden, The Netherlands. [5] Department of Medical Oncology, Amphia, P.O. Box 90157, 4800 RL Breda, The Netherlands. [6] Department of Medical Oncology, Haga hospital, P.O. Box 40551, 2504 LN Den Haag, The Netherlands. [7] Department of Medical Oncology, Viecuri, 5912BL Venlo, The Netherlands. [8] Department of Medical Oncology, Deventer hospital, P.O. Box 5001, 7416 SE Deventer, The Netherlands. [9] Department of Medical Oncology, Noordwest hospital group, location Alkmaar, P.O. Box 501, 1815 JD Alkmaar, The Netherlands. [10] Department of Medical Oncology, Hospital Gelderse vallei, 6710 HN Ede, The Netherlands. [11] Department of Surgery, Leiden University Medical Center, P.O. Box 9600, 2300 RC Leiden, The Netherlands. [12] Department of Medical Statistics and Bioinformatics, Leiden University Medical Center, P.O. Box 9600, 2300RC Leiden, The Netherlands. [13] Longevity Institute, School of Gerontology, and Department of Biological Sciences, University of Southern California, Los Angeles, CA 90089, USA. [14] IFOM FIRC Institute of Molecular Oncology, Via Adamello 16, Milan, Italy. [15] Department of Endocrinology, Leiden University Medical Center, P.O. Box 9600, 2300 RC Leiden, The Netherlands. [16] BOOG Study Center, P.O. Box 9236, 1006 AE Amsterdam, The Netherlands. *A list of authors and their affiliations appears at the end of the paper. ✉email: j.r.kroep@lumc.nl

Extensive preclinical evidence suggests that short-term fasting and fasting mimicking diets (FMDs) can protect healthy cells against the perils of a wide variety of stressors, including chemotherapy, simultaneously rendering cancer cells more vulnerable to chemotherapy and other therapies[1–5]. Essentially, fasting causes a switch in healthy cells from a proliferative state towards a maintenance and repair state. Malignant cells, in contrast, seem to be unable to enter this protective state because of oncoprotein activity, and therefore fail to adapt to nutrient scarce conditions. Instead, fasting deprives proliferating cancer cells of nutrients, growth and other factors, which renders them more sensitive to cancer therapy and increases cell death[1,3]. The phenomenon by which normal but not cancer cells become protected to toxins is termed differential stress resistance (DSR)[2,3] whereas the specific sensitization of cancer cells to stress is called Differential Stress Sensitization (DSS)[1,6].

Declines of plasma levels of insulin like growth factor-1 (IGF-1), insulin and glucose are among the mediators of the effects of fasting on cancer cells, as these factors can promote growth and prevent apoptosis[1–4,6,7]. Fasting periods of at least 48 h are required to induce a robust decrease in circulating glucose, IGF-1 and insulin levels[6,8]. A very low calorie, low protein FMD was developed for its ability to cause metabolic effects on various starvation response markers similar to those caused by water-only fasting, while reducing the burden associated with a water only fast[9,10].

Small clinical studies showed that fasting as an adjunct to chemotherapy is safe and well tolerated, while it may reduce its toxicity[11–14]. This multicentre, open label, randomized DIRECT study was designed to evaluate the impact of an FMD on toxicity as well as on the radiological and pathological response to chemotherapy for breast cancer.

## Results

**Patient characteristics.** From February 2014 to January 2018, 131 patients were randomized (see consort diagram, Fig. 1). One patient withdrew informed consent before starting with chemotherapy and one patient was ineligible because of liver metastases, which were diagnosed a day after randomization. Of the 129 patients, 65 received FMD as an adjunct to chemotherapy and 64 patients used their regular diet. Thirty patients received FEC-T chemotherapy and 99 AC-T. Patient characteristics were equally distributed between groups (Table 1 and Supplementary Table 2).

**Interim analysis.** Because the overall (both arms) pCR turned out to be significantly lower (11.7%) than anticipated (which would require the recruitment of twice as many participants to be able to detect the hypothesized pCR difference between both arms in a subsequent phase III study), in addition to the worse-than-expected compliance, the Data Safety Monitoring Board advised to dispense with the phase III study. Therefore, we here present the results of the phase II study.

**Compliance.** Fifty three out of 65 patients (81.5%) completed the first FMD cycle, whereas over 50% completed 2 FMD cycles, which could be sufficient to impact the tumor response to chemotherapy in view of the effects of only one or a few FMD cycles in enhancing the efficacy of chemotherapy in mice[15]. 22 out of 65 patients (33.8%) used the FMD for at least 4 cycles (all AC or FEC cycles), and 20.0% of the patients complied during all cycles of chemotherapy (Supplementary Table 3). The main reason for non-adherence to the FMD was dislike of distinct components of the diet, perhaps induced by chemotherapy. In the regular diet group, 5 (7.8%) patients were not compliant (they decided to fast during one or more cycles of chemotherapy).

**Intention to treat (ITT) analysis.** Data on toxicity are shown in Supplementary Table 4. Grade III/IV toxicity, scored during all cycles of chemotherapy, was not significantly different between the FMD group (75.4%) and the regular diet group (65.6%). No grade V toxicity occurred. The percentage of patients who discontinued chemotherapy did not significantly differ between groups (27.7% FMD vs 23.8% control, $P = 0.580$). Notably, while side effects were similar in both arms, patients in the FMD arm did not receive dexamethasone before the AC chemotherapy cycles.

The radiological response and pathological response according to Miller and Payne are shown in Fig. 2 and Supplementary Table 5. The overall pCR rate was 11.7% and did not differ between the two groups (10.8% in FMD group versus 12.7% in control group; OR 0.830, 95% CI 0.282–2.442, $P = 0.735$). Interestingly, the radiologically complete or partial response, as measured by MRI or ultrasound before surgery, occurred

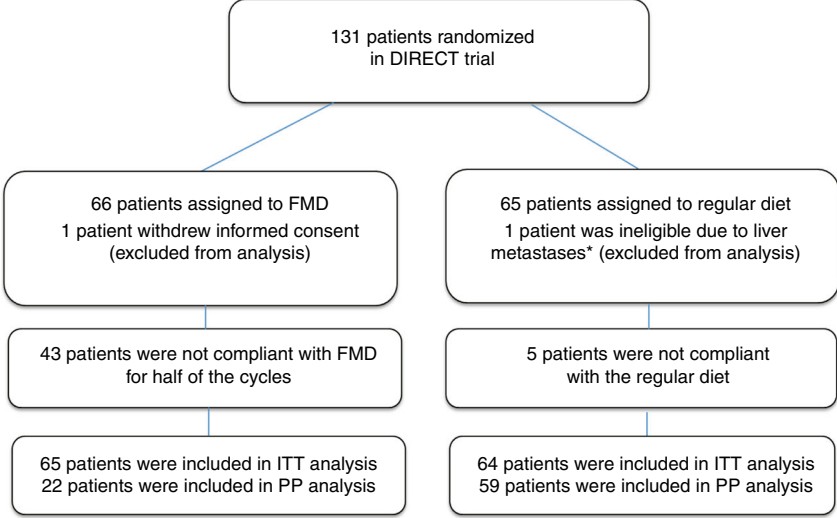

**Fig. 1 Consort diagram of the DIRECT study.** This figure shows reasons for exclusion from the study and the numbers of patients included in the PP and ITT analyses. Abbreviations: FMD: fasting mimicking diet, ITT: Intention to treat, PP: Per protocol. * diagnosed the day after randomization.

**Table 1 Patient characteristics.**

| | FMD (N = 65) | Regular diet (N = 64) |
|---|---|---|
| Median age (range), Years | 49.0 (31–71) | 51.0 (27–71) |
| Median body mass index (range), kg/m$^2$ | 25.7 (19.8–41.2) | 26.0 (19.7–39.0) |
| WHO status | | |
| Grade 0 | 61 (93.8%) | 60 (93.8%) |
| Grade 1 | 3 (4.6%) | 4 (6.3%) |
| Unknown | 1 (1.5%) | 0 (0%) |
| Menopausal status | | |
| Pre/Peri | 27 (41.5%) | 31 (48.4%) |
| Post | 38 (58.5%) | 31 (48.4%) |
| Unknown | 0 (0%) | 2 (3.1%) |
| T-classification | | |
| T1 | 5 (7.7%) | 6 (9.4%) |
| T2 | 42 (64.6%) | 41 (64.1%) |
| T3 | 17 (26.2%) | 15 (23.4%) |
| T4 | 1 (1.5%) | 2 (3.1%) |
| N-classification | | |
| N0 | 29 (44.6%) | 33 (51.6%) |
| N1 | 28 (43.1%) | 26 (40.6%) |
| N2 | 7 (10.8%) | 4 (6.3%) |
| N3 | 1 (1.5%) | 1 (1.6%) |
| Stage | | |
| I (ineligible) | 0 (0%) | 1 (1.6%) |
| II | 51 (78.5%) | 48 (75.0%) |
| III | 14 (21.5%) | 15 (23.4%) |
| HR status | | |
| ER−/PR− | 14 (21.5%) | 7 (10.9%) |
| ER-/PR unknown | 0 (0%) | 1 (1.6%) |
| ER+/PR− | 9 (13.8%) | 9 (14.1%) |
| ER+/PR+ | 42 (64.6%) | 47 (73.4%) |
| Chemotherapy regimen | | |
| AC-T | 52 (80.0%) | 47 (73.4%) |
| FEC-T | 13 (20.0%) | 17 (26.6%) |
| Grade (BR) | | |
| I | 2 (3.1%) | 2 (3.1%) |
| II | 43 (66.2%) | 42 (65.6%) |
| III | 20 (30.8%) | 19 (29.7%) |
| Unknown | 0 (0%) | 1 (1.6%) |
| Tumor type | | |
| Ductal | 53 (81.5%) | 49 (76.6%) |
| Lobular | 9 (13.8%) | 13 (20.3%) |
| Other | 3 (4.6%) | 2 (3.1%) |

*FMD* Fasting mimicking diet, *HR* hormone receptor, *AC-T* doxorubicin/cyclophosphamide followed by docetaxel, *FEC-T* Fluorouracil/epirubicin/cyclophosphamide followed by docetaxel, *BR* Bloom Richardson, *ER* estrogen receptor, *PR* progesterone receptor.

approximately 3 times more often in the FMD group compared to the control group in univariate (OR 2.886, 95% CI 1.012–8.227, $P = 0.047$) and multivariate (OR 3.168, 95% CI 1.062–9.446, $P = 0.039$) analyses. Accordingly, the proportion of patients with stable or progressive disease was 2.5 fold lower in the FMD group (11.3%) than in the control group (26.9%, Fig. 2).

The FMD affected various metabolic and endocrine parameters in the ITT analysis (Supplementary Table 6). At day −1/ 0 pre-chemotherapy, plasma insulin was significantly lower in the FMD group ($P = 0.004$), while a trend for lower plasma glucose levels was observed in the FMD group ($P = 0.062$). Urine ketone bodies were higher in the FMD group versus the control group ($P < 0.0001$).

Data on global QoL and the distress thermometer are shown in Fig. 3 and Supplementary Fig. 2, respectively. QoL was not significantly different between both groups in terms of global QoL ($P = 0.841$) and overall distress ($P = 0.674$).

**Per protocol (PP) analysis**. A PP analysis was done to substantiate the effects of FMD on toxicity and efficacy of chemotherapy. Specifically, patients who were compliant with the FMD for at least half of the cycles were compared with those who were less compliant, and with the compliant control patients (i.e., the patients in the control group who did not fast on their own initiative).

Toxicity data of the PP analysis are shown in Table 2. Grade III/IV toxicity did not differ between FMD compliant patients ($n = 22$) *vs.* control ($n = 59$) group.

In the PP analysis, the pCR rate did not differ between the compliant FMD patients (13.6%) and controls (12.1%, OR 1.150, 95% CI 0.269–4.911, $P = 0.850$, Supplementary Table 5). However, the Miller and Payne pathological response 4/5 (90–100% tumor cell loss) occurred more often in patients using FMD in both univariate (OR 3.194, 95% CI 1.115–9.152, $P = 0.031$) and multivariate analyses (OR 4.109, 95% CI 1.297–13.02, $P = 0.016$, Fig. 2) than in the control group. Furthermore, the more FMD cycles completed, the more patients had either a complete or partial radiological response to therapy ($P$ for trend = 0.035, Fig. 4). Both analyses were adjusted for hormone receptor status, TNM stage, BMI and chemotherapy regimen.

In the PP analysis (Fig. 5 and Supplementary Table 5), glucose was significantly lower in the compliant FMD group compared with the regular diet group before the first cycle and halfway therapy ($P = 0.006$ and $P = 0.042$, respectively). Insulin was significantly lower in the compliant FMD group compared with the control group before the first cycle and halfway therapy ($P = 0.001$ and $P < 0.001$, respectively). IGF-1 was significantly lower halfway therapy in patients who were compliant to the FMD in comparison to control patients ($P = 0.025$). Ketone bodies were positive in most of the compliant FMD patients (93.3%) and rarely positive in the compliant control group (8.1%, $P < 0.0001$).

The level of γ-H2AX intensity are reported in Supplementary Table 6. Only compliant patients were included. γ-H2AX intensity increased 3 h after chemotherapy in both groups for each cell type due to chemotherapy. The increase in DNA damage after chemotherapy was significantly less in CD45+ CD3+ T-lymphocytes from patients who had FMD as compared to patients using regular diet ($P = 0.045$, Fig. 6).

## Discussion

This is the first randomized controlled study evaluating the effects of an FMD on toxicity and efficacy of chemotherapy in patients with cancer. The results suggest that an FMD significantly reinforces the effects of neoadjuvant chemotherapy on the radiological and pathological tumor response in patients with HER2 negative early breast cancer. The ITT analysis reveals an increase of patients with a radiologically complete or partial response and a reduction of patients with radiologically stable/progressing disease in the FMD group compared to the control group. The PP analysis shows a beneficial effect of the FMD on the pathological response according to Miller and Payne. The more cycles of FMD were adhered to, the higher the percentage of Miller and Payne scores 4/5 (documenting >90% tumor cell loss) in the surgical specimen (Fig. 4).

By chance, the percentage of patients with a triple negative tumor randomized to receive the FMD was double the percentage of those in the control group (Table 1). pCR is more likely to occur in case of triple negative tumors[16,17]. However, triple negative tumors were significantly less common in patients who complied with the FMD than in those who did not, while the response of the tumor to chemotherapy was clearly more favorable in compliant patients (Supplementary table 2). Moreover, the positive effects of the FMD persisted after adjustment for the

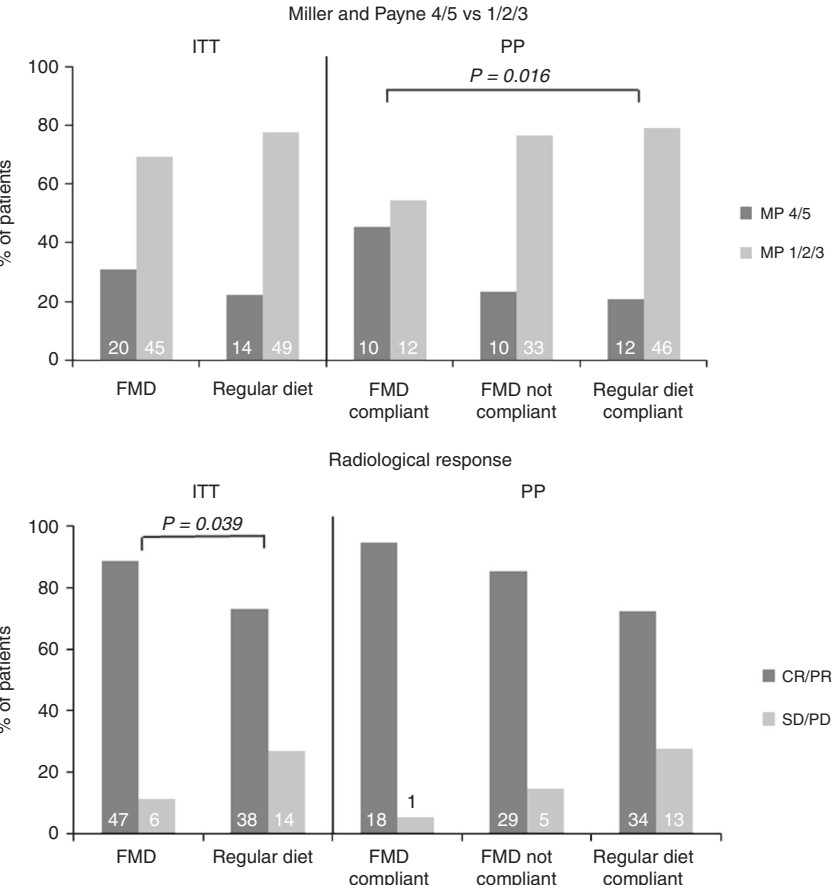

**Fig. 2 Tumor response data for the ITT and PP analysis.** The pathological response was given for Miller and Payne pathological response score 4/5 (90–100% tumor cell loss) vs. 1/2/3 (less than 90% tumor cell loss). The radiological response was scored according RECIST 1.1 and given for complete response + partial response vs. stable disease + progression disease. Abbreviations: FMD: fasting mimicking diet, ITT: Intention to treat, PP: Per protocol, MP: Miller and Payne, CR: complete response, PR: partial response, SD: stable disease, PD: progression disease. Logistic regression was used (2-sided). Source data are provided as a Source Data file.

receptor status of the tumor. These facts suggest that it was the FMD rather than the hormone receptor status which determined the better response of the tumor in agreement with the extensive pre-clinical data.

Patients using the FMD as an adjunct to chemotherapy did not experience more grade III/IV adverse events than patients who did not follow a diet, despite the fact that they were not prescribed dexamethasone in concert with FEC/AC. This suggests that the FMD may obviate the need for dexamethasone in the prevention of the side effects of chemotherapy. Importantly, DNA damage in T-lymphocytes was less in patients who received the FMD in combination with chemotherapy compared to those receiving chemotherapy while on a regular diet, suggesting that the FMD protected these cells against the induction of DNA damage by chemotherapy.

The study was meant to be a phase II/III study to evaluate the effects of the FMD on toxicity and efficacy of chemotherapy, respectively. However, a pre-defined interim analysis revealed a lower than anticipated overall pCR rate in the combined arms (albeit similar to the pCR rate in a similar trial of the same BOOG group[18]), necessitating the recruitment of twice as many participants to reliably judge the impact of the FMD on this primary outcome measure. Because this would prolong the study period and require additional funds, the DSMB advised to stop and report the results at the completion of phase II. Remarkably, the phase II study, involving only 131 patients, was sufficient to show

benefits of the FMD in sensitizing breast cancer cells to chemotherapy, with efficacy demonstrated both at the clinical and pathology levels.

Pre-clinical data, that has been accumulating for over 10 years, indicates that fasting can protect cancer-bearing mice against the side effects of chemotherapy[3], while sensitizing the tumor to its toxic effects[1,2]. Even one or a few cycles of FMD by itself can inhibit the progression of a wide variety of cancers and increase the therapeutic efficacy of chemotherapy in mice[1,15], but can also prime breast cancer and other tumor cell types to an attack by immune cells[15]. Accordingly, in spite of the fact that many patients could not adhere to the dietary regimen during all cycles of chemotherapy, our intention to treat analysis reveals benefits in terms of tumor response.

Only a few, generally small clinical studies have evaluated the potential of fasting to improve cancer treatment[11–14], primarily focusing on feasibility and toxicity of treatment. Just two of these trials were randomized[11–14], but the results were in line with those of the current study. Previously, we reported reduced hematological toxicity and DNA damage in circulating mononuclear cells in a small group of women who fasted for 24 h prior to (neo)adjuvant chemotherapy for breast cancer[11]. A second randomized study revealed improved QoL and less fatigue in breast- and ovarian cancer patients, who fasted for 60 h around the time of chemotherapy[14]. Yet another study reported a trend towards less grade 3–4 neutropenia and reduced DNA damage in

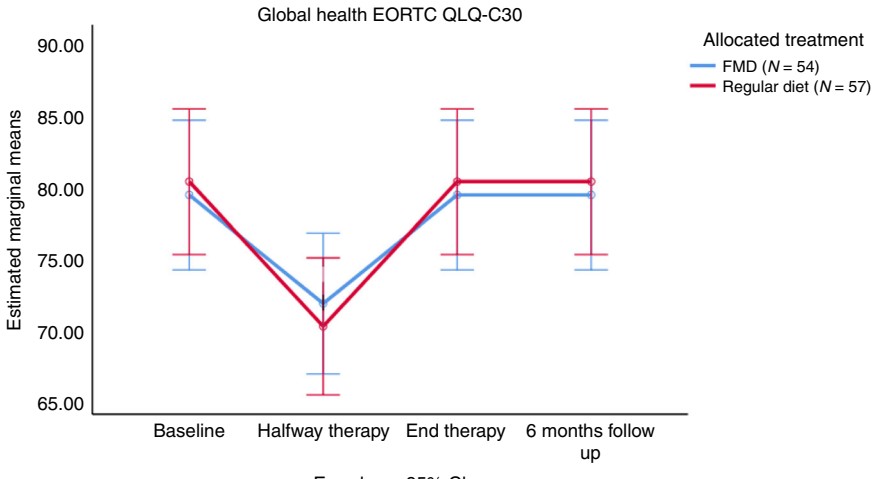

**Fig. 3 EORTC QLQ-C30 global health domain given for 4 timepoints: before therapy (after randomization), halfway therapy, at the end of therapy and at six months follow-up.** Error bars indicate the 95% CI of the mean. Abbreviations: EORTC: European Organization for Research and Treatment of Cancer, FMD: fasting mimicking diet. CI: confidence interval. FMD $N = 57$ and regular diet $N = 54$. Armitage's trend test was used. Source data are provided as a Source Data file.

**Table 2 Grade III/IV toxicity in both groups (ITT) and in patients who were compliant with the FMD for at least half cycles of CT vs. control patients who did not fast on their own initiative (PP).**

| Grade III/IV | FMD ($N = 65$) | FMD-C ($N = 22$) | FMD-NC ($N = 43$) | Control ($N = 64$) | P-value (ITT) | P-value (PP) |
|---|---|---|---|---|---|---|
| Total | 31 (47.7%) | 11 (50.0%) | 20 (46.5%) | 36 (56.3%) | 0.331 | 0.539 |
| Neutropenic fever | 5 (7.7%) | 1 (4.5%) | 4 (9.3%) | 5 (7.8%) | 0.980 | 0.548 |
| Neutropenia | 19 (29.2%) | 6 (27.3%) | 13 (30.2%) | 18 (28.1%) | 0.890 | 0.777 |

Grade III/IV side effects were scored according CTCAE4.03. Each side effect was scored maximal once per patient during the course. *FMD* fasting mimicking diet, *C* compliant, *NC* not compliant, *ITT* intention to treat, *PP* per protocol, *CT* chemotherapy.

leukocytes in patients who fasted for 48–72 as compared to 24 h around the time of platinum-based chemotherapy for a variety of malignancies[12]. Finally, fasting for variable time periods may reduce adverse events of chemotherapy, which was suggested by a case series of 10 patients with various cancer types[13]. These data agree with the current data, showing that the FMD is safe and effective as an adjunct to chemotherapy, at least in patients with a normal body mass index at inclusion.

Our data should be cautiously interpreted, particularly those of the PP analysis, which bears the risk of selection bias. However, the ITT analysis confirms the positive impact of the FMD on the radiological response, whilst clearly showing a trend in support of the PP positive effect on the pathological response. Moreover, due to self-selection bias patients in the control group decided to fast on their own initiative, which may have decreased the positive impact of the FMD in the ITT analysis.

In conclusion, the results of this study are the first to suggest that FMD cycles are safe and effective as an adjunct to chemotherapy in women with early breast cancer. These findings together with preclinical data encourage further exploration of the benefits of fasting/FMD in patients receiving a wide range of cancer therapies.

## Methods

**Study design and patients**. This is a randomized, controlled, observer-blind study. Eligible patients from 11 Dutch centers had histologically confirmed diagnosis of HER2-negative, stage II/III (cT1cN + or ≥T2 any cN, cM0) early breast cancer, adequate bone marrow reserve (white blood counts >3.0 × 10⁹/L, absolute neutrophil count ≥1.5 × 10⁹/L and platelet count ≥100 × 10⁹/L), adequate liver function (bilirubin ≤1.5 × upper limit of normal (UNL) range, ALAT and/or ASAT ≤2.5× UNL, Alkaline Phosphatase ≤5× UNL), adequate renal function (calculated creatinine clearance ≥50 mL min⁻¹), normal cardiac function, a WHO performance state 0–2, age ≥18 years, BMI >19 kg m⁻², absence of diabetes mellitus, absence of allergies for FMD content, and signed informed consent. The study (NCT02126449) was conducted in accordance with the Declaration of Helsinki (October 2013) and approved by the Ethics Committee of the Leiden University Medical Center in agreement with the Dutch law for medical research involving human subjects.

**Drugs**. Women received 8 cycles of neo-adjuvant AC-T chemotherapy (4 cycles doxorubicin 60 mg m⁻² and cyclophosphamide 600 mg m⁻² intravenously (i.v.)), followed by 4 cycles of T (docetaxel 100 mg m⁻² i.v.), or 6 cycles of neo-adjuvant FEC-T chemotherapy, consisting of 3 cycles of 5-fluorouracil, epirubicin and cyclophosphamide at a dose of 500, 100 and 500 mg m⁻² i.v., respectively), followed by 3 cycles of T (docetaxel 100 mg m⁻² i.v.), all q 3 weeks. The anti-emetic agents granisetron (1 mg i.v.) or ondansetron (8 mg i.v.) were administered prior to chemotherapy. Dexamethasone (8 mg i.v.) was administered shortly before chemotherapy for all cycles in the control group, whereas it was omitted during the AC or FEC courses in the FMD group, as dexamethasone may counteract the endocrine and metabolic effects of dietary intervention in the FMD group[19].

**Intervention**. Women were randomized in a 1:1 ratio to receive the FMD (Xentigen™) or regular diet for 3 days prior to and on the day of each cycle of chemotherapy. The FMD is a 4-day plant-based low amino-acid substitution diet, consisting of soups, broths, liquids and tea (Supplementary Fig. 3). Calorie content declined from day 1 (~1200 kcal), to days 2–4 (~200 kcal). Moreover, the carbohydrates/proteins/fats energy ratio was approximately 3.5/1/2 on the first day, while complex carbohydrates were the main macronutrient (>80 energy%) the other subsequent 3 days. Patients were allowed to eat the diet components at any time of the designated day.

**Randomization, masking and data storage**. Patients were centrally randomized at the LUMC datacenter through block randomization with various block sizes

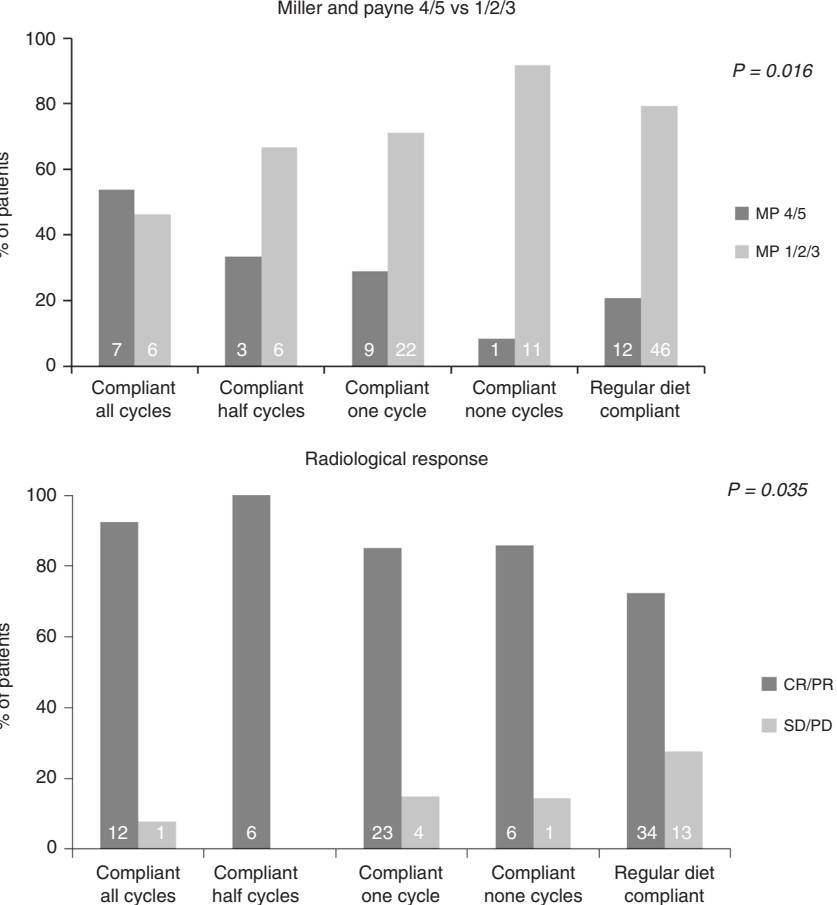

**Fig. 4 Tumor response data per number of cycles completed.** The pathological response was given for Miller and Payne pathological response score 4/5 (90–100% tumor cell loss) vs.1/2/3 (less than 90% tumor cell loss). The radiological response was scored according to RECIST 1.1 and given for complete response + partial response vs. stable disease + progression disease. *P*-value is given for Armitage's trend test (2-sided). Abbreviations: FMD: fasting mimicking diet, MP: Miller and Payne, CR: complete response, PR: partial response, SD: stable disease, PD: progression disease. Source data are provided as a Source Data file.

stratified by stage (II versus III), estrogen receptor status (positive versus negative), BMI ($<25 \text{ kg m}^{-2}$ versus $>25 \text{ kg m}^{-2}$) and chemotherapy regimen (AC-T versus FEC-T). The web based relational database management system ProMISe (https://www.msbi.nl/promise/ProMISe.aspx) was used for data storage and exchange.

**Blood sampling.** Venous blood samples were drawn prior to each chemotherapy administration (pre-chemotherapy on day −1 or day 0). Compliance with the diet was estimated by the following parameters: insulin, glucose, and IGF-1 (measured in a 9 mL serum-separating tube). All samples were analyzed by the accredited clinical laboratories of the participating centers.

The effect of FMD on chemotherapy-induced DNA damage in peripheral blood mononuclear cells (PBMCs) was examined in a side study. Sodium heparinized venous blood samples (9 mL) were collected for prior to the first cycle of chemotherapy and three hours after start of chemotherapy.

**Toxicity and efficacy.** The primary endpoint of the phase II and phase III parts of the study were grade III/IV toxicity and pathological complete response (pCR), respectively. Toxicity was documented by the physician and graded according to the Common Terminology Criteria for Adverse Events version4.03 (CTCAE v.4.03). Pathological complete response (pCR) was defined as the absence of residual invasive cancer within the breast and lymph nodes[16], excluding isolated tumor cells (ITC).

Secondary endpoints included radiological response and pathological response according to the Miller and Payne (Supplementary Table 1)[16]. Histopathology was centrally revised by one pathologist (DC), who was blinded to which treatment the patient received. Clinical response was measured by MRI or ultrasound of the breast halfway and at the end of therapy, according to RECIST1.1[20].

**Quality of life (QoL).** Global health was assessed with the EORTC QLQ-C30[21] before therapy (after randomization), halfway therapy, at the end of therapy and at

six months follow-up. Higher scores (0–100 scale) on the functional scales indicate a better QoL.

Psychosocial distress was measured with the distress thermometer[22], with an 11-point range from 0 (no distress) to 10 (extreme distress). Patients were asked to circle the number that best described the overall distress they experienced in the past week at 3 timepoints: halfway therapy, at the end of therapy and at six months follow-up.

**DNA damage: isolation of PBMCs and γ-H2AX staining.** PBMCs were isolated using Ficoll-Amidotrizoaat (Pharmacy LUMC) gradient centrifugation according to the standard operating procedure of the Medical Oncology department of LUMC. Isolated PBMCs were carefully resuspended and 3 times washed in PBS (B. Braun, Melsungen, Germany). Samples were fixed in 1.5% formaldehyde and permealized in ice-cold pure methanol. Cells were washed 3 times in staining buffer (PBS with 5% bovine serum albumin (BSA, Sigma, St Louis, USA)) and stained for 30 min on ice with anti-CD45-PerCP-Cy5.5 (BD Bioscience, Breda, the Netherlands), clone 2D1 anti-CD3-PE (BD, clone SK7), anti-CD14-AF700 (BD, clone M5E2), anti-CD15-PE CF594 (BD, clone W6D3) and anti-γ-H2AX-AF488 (Biolegend, clone 2F3), followed by another washing step and resuspension in PBS. Per experiment we used 1,000,000 cells or more when available. The cell acquisition was performed immediately after the staining procedure on the flow cytometer (BD LSR Fortessa Flow Cytometer analyzer, BD Bioscience, Breda, The Netherlands) and data were analyzed using BD FACS Diva Software version 6.2. The CD45+ cells were gated, after which the CD3+ T-lymphocytes, CD3− non-T cells (also harboring B lymphocytes) or CD14+ CD15− monocytes were analyzed for the geomean (as measure for the intensity) of γ-H2AX (Supplementary Fig. 2).

**Statistical analysis.** The primary endpoint of phase II of the study was grade III/IV toxicity. Based on trials with similar neo-adjuvant chemotherapy[17,23,24], the statistical power analysis revealed that a total number of 128 patients (64 patients

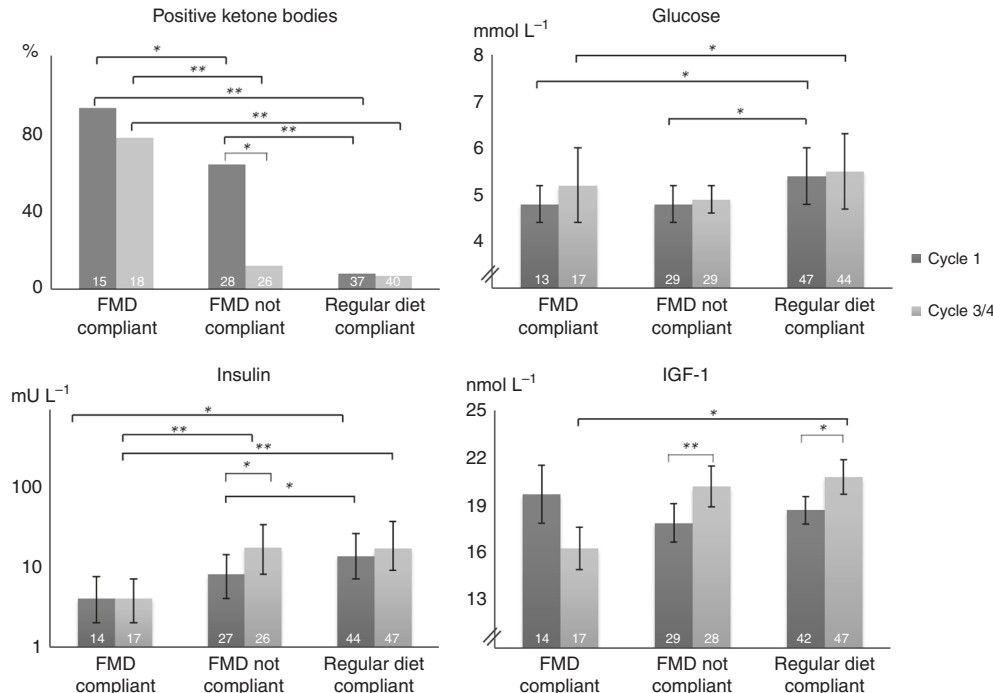

**Fig. 5 Metabolic and endocrine parameters before chemotherapy compared between compliant and non-compliant patients halfway therapy of the FMD group and the regular group.** Values are measured on day −1 or day 0 before cycle 1 and halfway therapy. *P-value <0.05, **P-value <0.001 (2-sided). Error bars indicate the standard error of the mean (if data was normally distributed) or the 25% and 75% percentiles of the median (if data was non-normally distributed). Independent t-tests and Mann-Whitney tests were used. Reference values: glucose 3.1–6.4 mmol/L; insulin 0–20 mU/L; IGF-1 5.4–24.3 nmol/L. Abbreviations: FMD: fasting mimicking diet, IGF-1: Insulin-like growth factor 1. Source data are provided as a Source Data file.

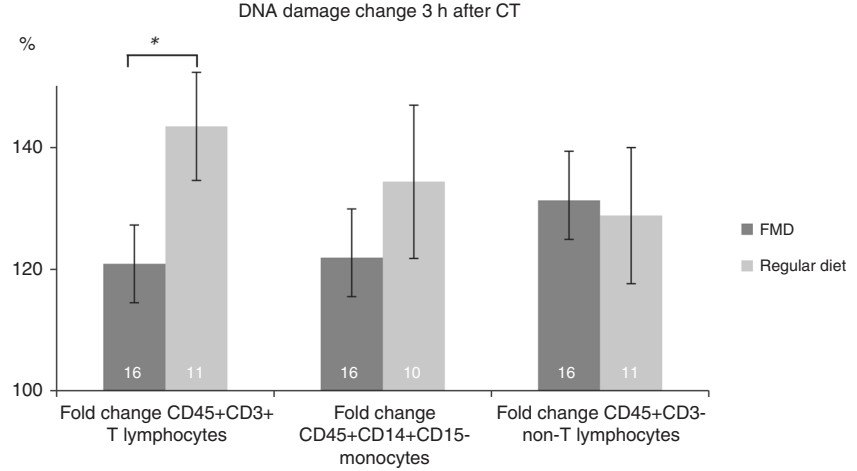

**Fig. 6 Difference of γ-H2AX intensity in CD45+ CD3+ lymphocytes of each patient before cycle 1 and 3 h after chemotherapy, given as a percentage increase.** Error bars indicate the standard error of the mean. Independent t-tests were used (2-sided). Abbreviations: FMD: fasting mimicking diet. *P = 0.045. Source data are provided as a Source Data file.

in each arm) was required to be able to detect a 50% reduction of grade III/IV adverse effects with 80% power using a nominal significance level of 3.06% .

The primary endpoint of the phase III part of the study was pathological complete response (pCR). We estimated the overall pCR rate to amount to 18%, based on studies examining similar third generation chemotherapy[17,18,23]. Our sample size calculation revealed that we would require a total number of 212 patients (106 per treatment arm).

An interim analysis, focusing on feasibility and adverse events, was planned after completion of the phase II part of the protocol by 128 patients and was approved by the Ethics Committee of the Leiden University Medical Center. Early stopping rules included significantly more or unacceptable adverse events in either group. A data safety monitoring board conducted the interim analysis. Survival data will be reported after 5 years follow-up.

All parameters were tested for normality using a Kolmogorov-Smirnov test, with Bonferroni adjustment when evaluated in subgroups. Normally distributed parameters, if necessary after log transformation, were summarized as mean (and standard error of the mean (SEM)) and compared using independent or paired samples t-tests when appropriate. The non-normally distributed parameters were summarized as median (and 25th and 75th percentiles) and compared using a Mann-Whitney test for independent groups or Wilcoxon signed rank test for paired groups. The effect of FMD on efficacy of chemotherapy was analyzed using logistic regression, yielding univariate and multivariate odds ratios (ORs), 95% confidence intervals (CIs), and P-values. Multivariate analyses were adjusted for stratification factors[25]. ER status, BMI, stage of disease and chemotherapy regimen. The Armitage's trend test was used to test an association between an ordinal variable and two categories. Mean changes in QoL from baseline to halfway, end of

therapy and 6 months follow-up were assessed in linear mixed models with 95% CIs. All tests were 2-tailed with a significance level of 0.05. All data were analyzed using IBM SPSS Statistics for Windows (Version 23.0. Armonk, NY: IBM Corp).

**Reporting summary**. Further information on research design is available in the Nature Research Reporting Summary linked to this article.

## Data availability

All study data are presented in the manuscript and supplementary materials. The source data underlying Tables 1 and 2, Figs. 2–6, Supplementary Fig. 1 and Supplementary Tables 2–7 are provided as a Source Data file. Additional raw data that support the findings of this study are available from the corresponding author upon reasonable request.

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

## Acknowledgements

We are greatly indebted to the patients for participating in this study, and their physicians for including the patients: E. Göker (Alexander Monro Hospital), A.J.M. Pas ('t Langeland Hospital) A.H. Honkoop (Isala). This work was supported by grants from Pink Ribbon (2012.WO31.C155) and Amgen (20139098).

## Author contributions

J.R.K., H.a.P., J.W.R., M.P.G.V., J.J.M.H., V.T.H.B.M. and V.D.L. contributed to study-concept/design, J.R.K., R.T.L., D.C., M.J.P.W., I.E., H.d.G., J.B.H., J.E.A.P., A.J.v.d.W., A.L.T.I., L.W.K., S.V., A.B., E.M.K.K., M.D.C. and S.d.G. contributed to data acquisition. H.e.P. and S.d.G. contributed to statistical analysis. S.d.G., H.a.P., J.R.K. and V.D.L. contributed to manuscript preparation. All authors contributed toward data analysis, drafting and revising the paper and agree to be accountable for all aspects of the work.

## Competing interests

V.D.L. has equity interest in L-Nutra. H.P. has shares in a company that invested in L-Nutra. The remaining authors declare no competing interests.

## Additional information

**Dutch Breast Cancer Research Group (BOOG)**

Hiltje de Graaf[4], Joan B. Heijns[5], Johanneke E. A. Portielje[1,6], Agnes J. van de Wouw[7], Alex L. T. Imholz[8], Lonneke W. Kessels[8], Suzan Vrijaldenhoven[9], Arnold Baars[10], Emine Göker[17], Anke J. M. Pas[18], Aafke H. Honkoop[19], A. Elise van Leeuwen-Stok[16] & Judith R. Kroep[1]

[17]Department of Medical Oncology, Alexander Monro Hospital, 3723 MB Bilthoven, The Netherlands. [18]Department of Medical Oncology, 't Langeland Hospital, 2725 NA Zoetermeer, The Netherlands. [19]Department of Medical Oncology, Isala hospital, 8025 AB Zwolle, The Netherlands.

