## [Peer Review File · Nature Communications]

Reviewers' Comments:

Reviewer #1:

Remarks to the Author:

De Groot et al conducted the first randomized controlled study evaluating the effects of a fasting mimicking diet (FMD) on toxicity and efficacy of chemotherapy in patients with HER2 negative early breast cancer. While no difference in toxicity was observed between treatment and control groups, FMD significantly enhanced the effects of neoadjuvant chemotherapy on radiological and pathological tumor response. This is an important study with novel biological and clinical insights. Comments include:

1. Summary: "the FMD significantly curtailed chemotherapy-induced DNA damage in lymphocytes" is not accurate. T or CD3+ cells should be used here instead of lymphocytes.
2. 22 out of 65 patients (33.8%) took FMD for at least 4 cycles. What were the characteristics of these patients as well as those of non-complied patients? In addition, can the authors comment on why a significant portion of patients could not adhere to the dietary regimen during chemotherapy – did FMD affect the health conditions of some patients?
3. Fig. 5 and sTable 7: why CD45+CD13- cells were considered myeloid cells? How about the DNA damage status for B cells, CD4+, and CD8+ cells? What were the cell number % of CD4+, CD8+, B, monocytic, and myeloid cells in FMD and control groups?
4. Does fasting or FMD alone affect cancer development?

Reviewer #2:

Remarks to the Author:

This is a straight-forward, generally well-written report of a clinical trial on fasting influences on radiologic and other response metrics in women undergoing neoadjuvant chemotherapy. Despite a DSMB recommendation to stop early because of an unexpectedly low event rate the trial was positive for a primary and several secondary endpoints. This work will be of general interest and be a nice addition to the developing literature on fasting influences on therapeutic sensitivity. I have only minor comments:

- * Line 85: this is far too firm a statement given our current state of knowledge around how fasting molecularly works, particularly in the context of malignant cells. The intro needs to be written much more clearly to show that these are reasonable speculations but by far not proven and fully accepted by our community (yet)
- * Much more detail needed on the precise FMD schedule used and experimental details and justifications here
- * In general reporting needs to be much more detail and follow standards (e.g. SPIRIT, CONSORT checklists and diagrams, etc.)
- * Line 186: some grammar problem in this sentence, hard to work out the intended meaning
- * Recommend performing an exploratory analysis of metabolic & endocrine features as a function of number of FMD cycles completed

Reviewer #3:

Remarks to the Author:

This manuscript describes the results of a randomized, controlled trial looking at a fast mimicking diet versus a control group, in patients receiving neo-adjuvant chemotherapy for breast cancer. The protocol accrual was stopped short of what was planned for the proposed phase 3 trial. The results did not show decreased toxicity (as opposed to what was expected) but more antitumor effect on the cancer was observed. Following critique is provided.

1. Is not clear whether the pathologist was blinded to which treatment arm patient has received. Ideally, the pathologist would have been blinded.
2. Table 1 illustrates that the fasting group had more ER negative/PR negative cancers than the control group. ER negative and PR negative cancers are more likely to have a complete pathologic response, which was seen in that group. This should be stated more clearly in the manuscript, as this situation is likely not going to be appreciated by many readers of this manuscript.
3. Much of the discussion section repeats data that were presented in the results section, without actually discussing such.
4. While the data are "supportive" of the concept that the fasting group had more anticancer activity, this reviewer believes that the authors are making statements which are too strong. Examples of this in just the first paragraph of the discussion section are as follows: "shows that an FMD significantly reinforces the effects"; "demonstrates a substantial increase"; and "revealed clearly beneficial effects". All of these could be softened by statements such as: "seems to be associated with" or "may increase".
5. In a similar fashion, the second to the last sentence in the second to the last paragraph claims (regarding another manuscript that is referenced) that "fasting for variable time periods reduced adverse events of chemotherapy in a case series of 6 patients with various cancer types." While I did not look up this manuscript, I'm convinced that I would not be certain that there was a clear cause and effect resulting from fasting in that study.

Dear reviewers,

Our point-by-point response to the concerns:

Changes that are made in the manuscript in response to the review of reviewer 1:

1. Summary: “ the FMD significantly curtailed chemotherapy-induced DNA damage in lymphocytes” is not accurate. T or CD3+ cells should be used here instead of lymphocytes
This sentence has been changed in: “Also, the FMD significantly curtails chemotherapy-induced DNA damage in T-lymphocytes”(Line 74-75)

2. 22 out of 65 patients (33.8%) took FMD for at least 4 cycles. What were the characteristics of these patients as well as those of non-complied patients?
This is shown in supplementary table 2.

In addition, can the authors comment on why a significant portion of patients could not adhere to the dietary regimen during chemotherapy – did FMD affect the health conditions of some patients?

On page 5 is the main reason described of non-adherence to the FMD: “The main reasons for non-adherence to the FMD was dislike to of distinct components”(line135). An overview of reasons for non-adherence is given in supplementary table 3. The health condition of patients was not one of them.

3. Fig. 5 and sTable 7: why CD45+CD13- cells were considered myeloid cells? How about the DNA damage status for B cells, CD4+, and CD8+ cells? What were the cell number % of CD4+, CD8+, B, monocytic, and myeloid cells in FMD and control groups?

Your point is well taken. Unfortunately, we did not use an antibody directed against CD19 or CD20. The study was not designed to in-depth characterize the immune status of patients. T-cells were studied primarily to map the impact of chemotherapy on DNA of healthy cells exposed. We rephrased this population as non-T cells instead of myeloid cells. (Figure 6)

4. Does fasting or FMD alone affect cancer development?

This is unknown for humans. However, cycles of FMD were shown to cause a major reduction in cancer development in mice.

Because the DIRECT trial studied the effect of short term fasting on chemotherapy and not on cancer development we did not add this in the current paper.

Changes that are made in the manuscript in response to the review of reviewer 2:

Reviewer #2 (Remarks to the Author):

This is a straight-forward, generally well-written report of a clinical trial on fasting influences on radiologic and other response metrics in women undergoing neoadjuvant chemotherapy. Despite a DSMB recommendation to stop early because of an unexpectedly low event rate the trial was positive for a primary and several secondary endpoints. This work will be of general interest and be a nice addition to the developing literature on fasting influences on therapeutic sensitivity. I have only minor comments:

* Line 85: this is far too firm a statement given our current state of knowledge around how fasting molecularly works, particularly in the context of malignant cells. The intro needs to be written much more clearly to show that these are reasonable speculations but by far not proven and fully accepted by our community (yet)

Your point is well taken. Sentence 85 is rephrased in: *“Extensive preclinical evidence suggests that short-term fasting and fasting mimicking diets (FMDs) can protect healthy cells against the perils of a wide variety of stressors, including chemotherapy, simultaneously rendering cancer cells more vulnerable to chemotherapy and other therapies”* (Line 84-87)

Line 89 is rephrased in: *Malignant cells, in contrast, seem to be unable to enter this protective state because of oncoprotein activity, and therefore fail to adapt to nutrient scarce conditions.*

“Increasing safety and” is deleted (line 102).

* Much more detail needed on the precise FMD schedule used and experimental details and justifications here

The following information has been added (line 313-318):

-..., consisting of soups, broths, liquids and tea... ”

- ... subsequent...

- “Patients were allowed to eat the diet components at any time of the designated day.”

Moreover, in supplementary table 3 we added the exact schedule in a figure.

* In general reporting needs to be much more detail and follow standards (e.g. SPIRIT, CONSORT checklists and diagrams, etc.)

- *Inclusion and exclusion criteria are added in the summary*

- *A Consort diagram is added (Figure 1, line 112)*

* Line 186: some grammar problem in this sentence, hard to work out the intended meaning

The sentence is rephrased in: *“IGF-1 was significantly lower halfway therapy in patients who were compliant to the FMD in comparison to control patients”*. (line 197)

* Recommend performing an exploratory analysis of metabolic & endocrine features as a function of number of FMD cycles completed

In figure 5 metabolic and endocrine values are given for compliant patients (defined as compliant to half of the cycles of FMD) and not compliant patients at cycle 1 and halfway therapy.

Changes that are made in the manuscript in response to the review of reviewer 3:

Reviewer #3 (Remarks to the Author):

This manuscript describes the results of a randomized, controlled trial looking at a fast mimicking diet versus a control group, in patients receiving neo-adjuvant chemotherapy for breast cancer. The protocol accrual was stopped short of what was planned for the proposed phase 3 trial. The results did not show decreased toxicity (as opposed to what was expected) but more antitumor effect on the cancer was observed. Following critique is provided.

1. Is not clear whether the pathologist was blinded to which treatment arm patient has received. Ideally, the pathologist would have been blinded.

Yes, the pathologist was blinded. This is a very important point indeed. We added a separate paragraph: “*Study design and patients*” explicitly stating that it was an observer-blinded study. (line 287). Also, the following sentence was added: “*who was blinded to which treatment the patient received*”. (Line 344).

2. Table 1 illustrates that the fasting group had more ER negative/PR negative cancers than the control group. ER negative and PR negative cancers are more likely to have a complete pathologic response, which was seen in that group. This should be stated more clearly in the manuscript, as this situation is likely not going to be appreciated by many readers of this manuscript .

Your point is well taken. We mention this issue in the discussion now.

“*By chance, the percentage of patients with a triple negative tumor randomized to receive the FMD was double the percentage of those in the control group (Table 1). pCR is more likely to occur in case of triple negative tumors 19,24. However, triple negative tumors were significantly less common in patients who complied with the FMD than in those who did not, while the response of the tumor to chemotherapy was clearly more favourable in compliant patients (Supplementary table 2). Moreover, the positive effects of the FMD persisted after adjustment for the receptor status of the tumor. These facts suggest that it was the FMD rather than the hormone receptor status which determined the better response of the tumor in agreement with the extensive pre-clinical data..* (Line 223-231).

We also added the following sentence to the results

“*Both analyses were adjusted for hormone receptor status, TNM-stage, BMI and chemotherapy regimen.*“ (Line 189-190).

3. Much of the discussion section repeats data that were presented in the results section, without actually discussing such.

See answers to question 2 and 4.

4. While the data are “supportive” of the concept that the fasting group had more anticancer activity, this reviewer believes that the authors are making statements which are

too strong. Examples of this in just the first paragraph of the discussion section are as follows: “shows that an FMD significantly reinforces the effects”; “demonstrates a substantial increase”; and “revealed clearly beneficial effects”. All of these could be softened by statements such as: “seems to be associated with” or “may increase”.

The first paragraph of the discussion has been rephrased:

“The results suggest that an FMD significantly reinforces the effects of neoadjuvant chemotherapy on the radiological and pathological tumor response in patients with HER2 negative early breast cancer. The ITT analysis reveals an increase of patients with a radiologically objective response and a reduction of patients with radiologically stable/progressing disease in the FMD group compared to the control group. The PP analysis shows a beneficial effect of the FMD on the pathological response according to Miller and Payne. The more cycles of FMD were adhered to, the higher the percentage of Miller and Payne scores 4/5 (documenting > 90% tumor cell loss) in the surgical specimen (Fig. 2).” (line 222-230).

5. In a similar fashion, the second to the last sentence in the second to the last paragraph claims (regarding another manuscript that is referenced) that “fasting for variable time periods reduced adverse events of chemotherapy in a case series of 6 patients with various cancer types.” While I did not look up this manuscript, I'm convinced that I would not be certain that there was a clear cause and effect resulting from fasting in that study.

The sentence has been changed in: *“Finally, fasting for variable time periods may reduce adverse events of chemotherapy, which was suggested by a case series of 10 patients with various cancer types” (line 271-272).*

Thanking you for your consideration, we remain on behalf of all authors,

Reviewers' Comments:

Reviewer #1:

Remarks to the Author:

The authors have addressed my concerns.